# Complications following surgeries for endometriosis: A systematic review protocol

**Foruzan Bahrami[1], Sarah Maheux-Lacroix[1,2], Olga Bougie[3], Amélie Boutin[1,4,5] ***

1 Reproduction, Mother and Youth Health Unit, CHU de Québec-Université Laval Research Center, Quebec City, Quebec, Canada, 2 Department of Obstetrics and Gynaecology, Université Laval, Quebec City, Quebec, Canada, 3 Department of Obstetrics and Gynecology, Queen's University, Kingston, Ontario, Canada, 4 Population Health and Optimal Health Practices Unit, CHU de Québec-Université Laval Research Center, Quebec City, Quebec, Canada, 5 Department of Pediatrics, Université Laval, Quebec City, Quebec, Canada

* amelie.boutin@fmed.ulaval.ca

**Data Availability Statement:** No datasets were generated or analyzed during the preparation of the protocol.

## Abstract

### Background

Endometriosis is a common gynecological condition with a wide range of symptoms, including infertility, dyspareunia, intestinal disorders, and pelvic pain. Laparoscopy and laparotomy are used widely for diagnosing and managing endometriosis. We will conduct a systematic review and meta-analysis with the aims of reporting complications rates following each type of surgeries for endometriosis and determinants of complications.

### Method

We will search Medline (via PubMed), Embase, the Cochrane Library, Web of Science, and Google Scholar for both retrospective and prospective cohorts or trials of at least 30 participants reporting perioperative and postoperative complications for endometriosis surgeries. We will restrict the studies to those conducted after 2011, to be representative of current practices, and will exclude studies of surgeries for gynecological cancer, or other concomitant benign gynecologic surgeries such as myomectomy. Two reviewers will independently screen references and select eligible studies. A standardized form will be used to collect data related to the baseline characteristics, potential determinants of complications, types of interventions, and outcomes. Cumulative incidences of complications will be pooled using DerSimonian and Laird random-effects method. The relation between potential determinants and complications will be reported with risk ratios and their 95% of confidence intervals. Subgroup analysis of surgical approach, surgical procedure, superficial and deep infiltrating endometriosis, and the indication of surgery will be conducted. Sensitivity analyses restricted to studies with low risk of bias will be performed.

### Discussion

This systematic review will provide information on the rates of complications for different surgical approaches and procedures for the treatment of endometriosis. It will contribute to inform patients when making decisions regarding their care. Identifying potential

**Funding:** The author(s) received no specific funding for this work.

**Competing interests:** The authors have no conflict of interest to declare.

determinants of complications will also help to improve care by identifying women being at higher risk of complications.

## Trial registration

**Systematic review registration:** CRD42021293865.

## Introduction

Endometriosis is a chronic, debilitating gynecological disease caused by the presence of endometrial glands and stroma tissue outside the uterus and affects 5–10% of women of reproductive age [1–3]. It is associated with infertility, dyspareunia, intestinal disorders, and pelvic pain [4].

Both pharmacological and surgical therapies are used for the treatment of endometriosis [5]. Among the surgical approaches, laparoscopy is favored over laparotomy due to better visualization of endometriosis lesions, as well as shorter hospitalization and return to daily activities [4]. Peri- and post-operative complications of endometriosis surgeries include bleeding, infection, thrombophlebitis, pulmonary embolism, and injury to the bowel or urinary tract system [6–9]. An approach by laparotomy or conversion from laparoscopy to laparotomy and a more invasive surgical procedure such as hysterectomy, concomitant intestinal or bladder resection, ureteric surgery increases the complication rates [10]. Prior studies have reported postoperative complications rates ranging from 0.4% to 4.7% [6–9, 11]. In addition, iatrogenic pelvic organ dysfunction can happen due to accidental damage to pelvic nerves in complete excision of deep-infiltrating endometriosis despite practices in nerve-sparing surgeries [12, 13]. The severity of the extent of lesions in the pelvis, usually reported using the r-ASRM staging system and ENZIAN classification, has been reported as directly related to both perioperative and postoperative complications [6–8, 11]. High level of care centers, expertise of the surgery team, pervious surgery for endometriosis, and complex procedures such as segmental resection and urinary tract involvement surgeries have also been postulated as possible determinants of the risk of perioperative and postoperative complications [6–8, 11, 14–16]. However, perioperative and postoperative complication rates following different types of endometriosis-related surgeries remain unclear. Furthermore, data on potential determinants of the risk of complications are scarce. We, thus, aim to describe the complication rate of all types of endometrioses-related surgeries in cohort studies and their determinants.

## Materials and methods

We will conduct a systematic review of cohort studies and trials reporting the complications rate and potential determinants of the complications of endometrioses surgeries. The protocol of our review is registered in PROSPERO (CRD42021293865). The review design follows the methodological recommendations of the Cochrane Handbook for the Systematic Reviews of Interventions [17] and is reported according to the Preferred Reporting Items for Systematic Reviews and Meta-Analysis (PRISMA) statement [18].

### Eligibility criteria

Prospective and retrospective cohort studies or trials of surgeries for managing endometriosis including at least 30 participants and reporting perioperative and/or postoperative

**Table 1. Eligibility criteria.**

| | |
|---|---|
| Population | Female patients with any types of endometriosis, excluding cases with concomitant gynecological cancer diagnosis, or other |
| Intervention | All types of surgeries for managing endometriosis, excluding cases with concomitant benign gynecologic surgeries such as myomectomy |
| Outcome | Perioperative and postoperative complications |
| Study Design | Prospective and retrospective cohort studies or trials of endometriosis surgeries including at least 30 participants conducted after 2011 |

complications will be eligible (Table 1). We will include studies conducted after 2011 for our review to be representative of current or recent practices. Selection of recent studies will allow to focus on current practices. There will be no restriction on language. Studies of patients with a gynecological cancer diagnosis, or other concomitant benign gynecologic surgeries such as myomectomy will be excluded.

## Information sources

We will search Medline (via PubMed), Embase, the Cochrane Library, Web of Science, and Google Scholar to identify all eligible studies. References of the selected studies and conferences abstract will be checked to find other eligible studies. Any additional potentially eligible studies identified by gynecological experts in our team will be screened. References will be exported in the online software Covidence [19], and duplicates will be removed using this tool.

## Search strategy

The searched terms include endometriosis, full range of endometriosis surgeries, and complications. An information specialist and a gynecology expert in our team were consulted to ensure the completeness of our search strategy. A validated filter will be used to exclude any non-human animal studies [17]. The full search strategy for Medline is available in S1 Appendix.

## Study selection

The references will be screened by an author, first based on the titles and abstracts. Thereafter, full texts of references deemed potentially eligible will be assessed for inclusion by 2 reviewers independently. If there is any disagreement, a third reviewer will be involved. All potentially eligible articles in languages other than English will be translated to English for eligibility assessment. A PRISMA flow diagram will be created to document the excluded references and reasons for full-text exclusions.

## Data collection process

Data extraction will be conducted by two independent reviewers. A standardized extraction form (S2 Appendix) will be piloted on 3 studies, after which adjustments will be made as needed. A third reviewer will be consulted if any disagreements arise. The authors of original studies will be contacted in case of ambiguity in the reported data.

## Data items

We will extract the characteristics of the study (year, location, design, sample size, follow-up duration), the characteristics of the participants, and potential determinants of surgical complications (age, comorbidities, BMI, surgical history, pervious surgery for endometriosis,

parity, symptoms, race/ethnicity, severity of endometriosis, level of care of the facility, and expertise of the surgery team), the characteristics of the intervention (type of route for the surgeries, the type of procedures, and indication), and outcomes (any complications reported by authors, number of participants with any postoperative complications, number of participants with major and minor complications, follow-up duration, along with moment of measurement of the outcome).

## Risk of bias in individual studies

The Cochrane collaboration tool for the assessment of the "Risk Of Bias in Non-randomized Studies of Interventions" (ROBINS-I) will be used to assess the risk of bias in individual studies [20]. Two independent reviewers will apply the tool to each selected study, with a third reviewer assisting with any disagreements.

## Outcomes

Our primary outcome will be a composite outcome of any intraoperative and postoperative complications (including conditions such as hemorrhage; organ injury; nerve damages including urinary voiding dysfunction, and urinary retention; open surgery conversion; mortality; postoperative reintervention; readmission after discharge from hospital; length of surgery; and length of hospital stay). The full list of complications is provided in the S2 Appendix. In addition, We will categorize complications as intraoperative complications, which occur during surgery, and postoperative complications, which occur after surgery. Postoperative complications will be reported within 30 days and three months after the surgery. We will also consider specific complications, and determinants of complications as secondary outcomes.

## Summary measures

The frequency of the complications will be reported by the cumulative incidence in hospital, at 30 days and at three months (any other time point may be considered if a sufficient number of studies report complications within this period) along with 95% confidence intervals (CI). Risk ratio and their 95% CI will be used to assess the relationship between complications and potential determinants (categorical: age, BMI, surgical history, previous surgery for endometriosis, complex procedures such as segmental resection, parity, race/ethnicity, high level of care centers, and expertise of the surgery team, severity). The severity of endometriosis will be classified according to the revised American Society for Reproductive Medicine (rASRM) or/and ENZIAN. In addition, the level of care will be classified as rural/community/regional hospitals versus academic/national/specialized hospitals versus unknown or mixed hospitals. When surgeon expertise is available in the included articles, we will extract it. Since there is no official classification, we will create categories if necessary after consulting with clinicians and method experts in our group, unless we report it as a part of descriptive analyses. Furthermore, the Clavien-Dindo classification is used to classify major and minor complications (grades I-II: minor complications, grades III-IV: major complications, grade V: death). This classification determines the severity of complications by analyzing the therapy used to treat them. If necessary, we will consult a gynecological expert in our team. Moreover, if adjusted measures of association are reported, we will collect the measure of association from the original studies and list of adjustment factors.

## Synthesis of the results

Laparotomy and laparoscopy methods will be analyzed separately. We will use the DerSimonian and Laird random-effects approach to pool proportions using R statistical software

(RStudio Version 1.2.5001, © 2009–2019 RStudio, Inc.). Crude risk ratios will be pooled using the Mantel-Haenszel method with random-effect models. Adjusted risk ratios will be pooled using generic inverse variance with random-effect models. Pooling of measures of associations will be conducted in the Cochrane statistical package RecMan5 (version 5.1, Copenhagen: The Nordic Cochrane Centre, The Cochrane Collaboration, 2012). Heterogeneity will be assessed with the Cochran's Q test and $I^2$, and we will interpret $I^2$ as low from 0 to 40%, moderate from 30 to 60%, substantial from 50 to 90%, and considerable from 75 to100%.

We will conduct subgroup analyses by 1) surgical approach (laparotomy or laparoscopy methods), 2) surgical procedure (hysterectomy, complex surgical technique including bowel or urinary resection, concomitant multiple procedures, and shaving and disc excision), 3) type of endometriosis (superficial or deep/extensive endometriosis; stages 1–4), 4) indication for surgery (infertility, pain, other), 5) complications considered (only severe or any complications; perioperative or postoperative or any). The adenomyosis information also will be collected if it is available in the studies for subgroup analysis. We will conduct sensitivity analysis restricting to studies deemed at low risk of bias. A two-sided 5% type I error will be considered in all confidence intervals calculations and statistical tests.

## Risk of publication bias

The risk of publication bias will be assessed by using the visual exploration of funnel plots.

## Ethics

Ethics approval not required as the study will rely on published data.

## Discussion

### Expected benefits

Endometriosis is a challenging disease. Although pharmacological treatment is the first-line therapy for endometriosis, a significant proportion of patients undergo surgery. This systematic review and meta-analysis will focus on the complications of various endometriosis surgeries, allowing patients to make more informed healthcare decisions. In addition, identifying determinants of complications will enable surgeons to recognize women who are at higher risk of complications in order to optimize care. It will also provide evidence to inform future research on surgeries for endometriosis.

### Limitations

We do expect a high heterogeneity in endometriosis severity of patients included in studies. Studies will likely report composite outcomes for complications including different conditions. We plan to conduct subgroup analyses to address these sources of heterogeneity. When looking at the association between potential determinants and complications, we expect potential confounding bias. We will thus extract any adjusted measure of association reported by the authors.

Our review will be based on an exhaustive search strategy through various reference databases. It will therefore provide a synthesis of the evidence on complications of endometriosis surgeries, allowing to identify potential areas of improvement or populations at greater risk of complications. Thus, it will enable to identify gaps and avenues for future research to improve practices.

## Supporting information

**S1 Checklist. PRISMA-P 2015 checklist.**
(PDF)

**S1 Appendix. Search strategy for Medline (Ovid) performed on August 20, 2021.**
(PDF)

**S2 Appendix. Data extraction form.**
(PDF)

## Acknowledgments

The authors would like to thank Chantal Beauregard who provided her help in the development of the search strategy.

## Author Contributions

**Conceptualization:** Foruzan Bahrami, Sarah Maheux-Lacroix, Amélie Boutin.

**Formal analysis:** Foruzan Bahrami.

**Investigation:** Amélie Boutin.

**Supervision:** Sarah Maheux-Lacroix, Amélie Boutin.

**Writing – original draft:** Foruzan Bahrami, Amélie Boutin.

**Writing – review & editing:** Sarah Maheux-Lacroix, Olga Bougie, Amélie Boutin.

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
