## [Decision Letter · Decision Letter 0]

18 Oct 2022

PONE-D-22-21731

Complications following surgeries for endometriosis: a systematic review protocol

PLOS ONE

Dear Dr. Boutin,

Thank you for submitting your manuscript to PLOS ONE. After careful consideration, we feel that it has merit but does not fully meet PLOS ONE’s publication criteria as it currently stands. Therefore, we invite you to submit a revised version of the manuscript that addresses the points raised during the review process.

We look forward to receiving your revised manuscript.

Kind regards,

Diego Raimondo

Academic Editor

PLOS ONE

Journal Requirements:

Reviewers' comments:

Reviewer's Responses to Questions

**Comments to the Author**

1. Does the manuscript provide a valid rationale for the proposed study, with clearly identified and justified research questions?

Reviewer #1: Yes

Reviewer #2: Yes

2. Is the protocol technically sound and planned in a manner that will lead to a meaningful outcome and allow testing the stated hypotheses?

Reviewer #1: Partly

Reviewer #2: Yes

3. Is the methodology feasible and described in sufficient detail to allow the work to be replicable?

Reviewer #1: Yes

Reviewer #2: Yes

4. Have the authors described where all data underlying the findings will be made available when the study is complete?

Reviewer #1: No

Reviewer #2: Yes

5. Is the manuscript presented in an intelligible fashion and written in standard English?

Reviewer #1: Yes

Reviewer #2: Yes

6. Review Comments to the Author

You may also provide optional suggestions and comments to authors that they might find helpful in planning their study.

Reviewer #1: Complications after endometriosis surgery is a very important topic and any attempt to analyze their frequency and determinants is welcome. In this manuscript, authors present a protocol for studying complications rates following each type of surgeries for endometriosis and determinants of those complications.

There are several pitfalls that must be addressed before accepting it for publication:

1.- In the Search Strategy section, only 3 databases are included: MEDLINE (via PubMed), EMBASE, and the Cochrane Library. However other important electronic databases (such as Web of Science, Wanfang data, ClinicalTrials, NLM Gateway, Google Scholar, or BIOSIS preview) may give also important information. The authors should explain why them are not searching those database.

2.- As the review wi bell reported according to the Preferred Reporting Items for Systematic Review and Meta-Analysis format, the inclusion criteria should be sufficiently followed by PICOS (P: participants, I: intervention, C: comparison, O: outcomes, S: study design) format.

3.- The Cochrane Collaboration is more suitable for randomized controlled trials, and it is expected that all the included studies will be nonrandomized controlled trials [3]. Therefore, Downs and Black Tools (27 items with a total score of 29) are recommended for authors (SH Downs, N. Black. The feasibility of creating a checklist for the assessment of the methodological quality both of randomised and non-randomised studies of health care interventions. J Epidemiol Community Health: 1998; 52, 377-384)

4.-The authors should clarify and more clearly define some of the terms they will use to explain characteristics of the study, the participants, and the potential determinants of complications. For example, how they will define or classify “symptoms”: will it be pain? severity of pain? intestinal or urinary symptoms? infertility?. How they will define “severity of endometriosis”? by using ASRM classification? If yes, it is well known that this classification has a bad correlation with surgical complications. If they opt by using ENZIAN classification, how they will built the final ENZIAN score?. Finally, although other definitions should be also provided, one the most important missing definition is the core of this protocol: how the author will define “surgical complication” and how they will define “major and minor complications”?. Other important missing definitions are how they will define “level care of centers”, “expertise of the surgery team”… Furthermore, the authors should explain why they are not taking in account other confounding factors as the association with adenomyosis is or if they will consider as a complication of endometriosis surgery the impairment of ovarian function frequently seen after ovarian endometrioma stripping.

Reviewer #2: Dear

Thank you to giving me the chance to review this interesting protocol on a very important topic

for the management of endometriosis

In order to explain better to the readers the importance of some important factors influencing iatrogenic complications (ie ureteral stent, parametrial or vaginal mucosa

Involvement) related to deep infiltrating endometriosis surgery I suggest to include and briefly comments some relevant articles (eg doi 10.1002/ijgo.14089, 10.1002/ijgo.13959, 10.1111/aogs.13824).

I believe that is important to list exauatively all complications that will be assessed, both de novo functional than organic ones.

7. PLOS authors have the option to publish the peer review history of their article (what does this mean?). If published, this will include your full peer review and any attached files.

Reviewer #1: No

Reviewer #2: No

---

## [Author Response · Author response to Decision Letter 0]

20 Jan 2023

We would like to thank the reviewers’ thorough comments. We have provided answers below and highlighted the changes made in the manuscript with tracked edits.

Comments to the Author

Reviewer #1: Complications after endometriosis surgery is a very important topic and any attempt to analyze their frequency and determinants is welcome. In this manuscript, authors present a protocol for studying complications rates following each type of surgeries for endometriosis and determinants of those complications.

There are several pitfalls that must be addressed before accepting it for publication:

1.- In the Search Strategy section, only 3 databases are included: MEDLINE (via PubMed), EMBASE, and the Cochrane Library. However other important electronic databases (such as Web of Science, Wanfang data, ClinicalTrials, NLM Gateway, Google Scholar, or BIOSIS preview) may give also important information. The authors should explain why them are not searching those databases. 

ANSWER: We have added searches of Web of Science and Google Scholar. After removing the duplicates, we did not find any new studies. We corrected the protocol to reflect these additional searches. Considering our eligibility criteria, we estimate the inclusion of the databases listed and search of references of included articles will ensure a very high sensitivity of our search.

Line 36-37 & 114-115: “We will search Medline (via PubMed), Embase, the Cochrane Library, Web of Science, and Google Scholar to identify all eligible studies.”

2.- As the review will be reported according to the Preferred Reporting Items for Systematic Review and Meta-Analysis format, the inclusion criteria should be sufficiently followed by PICOS (P: participants, I: intervention, C: comparison, O: outcomes, S: study design) format. 

ANSWER: We changed the presentation of the eligibility criteria to make explicit the PICOS. See Table 1. Since the aim is descriptive, there will be no comparison of interventions. Hence, no comparison group is explicitly presented.

3.- The Cochrane Collaboration is more suitable for randomized controlled trials, and it is expected that all the included studies will be nonrandomized controlled trials [3]. Therefore, Downs and Black Tools (27 items with a total score of 29) are recommended for authors (SH Downs, N. Black. The feasibility of creating a checklist for the assessment of the methodological quality both of randomised and non-randomised studies of health care interventions. J Epidemiol Community Health: 1998; 52, 377-384) 

ANSWER: The Downs and Black tool is a checklist looking a study quality and includes items related to reporting quality for example. Tools which are aiming at assessing the risk of bias are a little different and more appropriate in our opinion to evaluate the internal validity of specific studies. As such, we preferred to use the ROBINS-I tool developed by the Cochrane Collaboration, not to be confused with the RoB2 (Cochrane risk-of-bias tool for randomized trials). The ROBINS-I was specifically developed for the assessment of the risk of bias of non-randomized studies. We clarified that the tool is for non-randomized studies (Sterne et al., 2019). 

REF: Sterne, J. A., Hernán, M. A., McAleenan, A., Reeves, B. C., & Higgins, J. P. (2019). Assessing risk of bias in a non-randomized study. In Cochrane Handbook for Systematic Reviews of Interventions (pp. 621-641). https://doi.org/https://doi.org/10.1002/9781119536604.ch25

Line 160-162: ”The Cochrane collaboration tool for the assessment of the “Risk Of Bias in Non-randomized Studies of Interventions” (ROBINS-I) will be used to assess the risk of bias in individual studies (20). “ 

4.-The authors should clarify and more clearly define some of the terms they will use to explain characteristics of the study, the participants, and the potential determinants of complications. For example, how they will define or classify “symptoms”: will it be pain? severity of pain? intestinal or urinary symptoms? infertility?. How they will define “severity of endometriosis”? by using ASRM classification? If yes, it is well known that this classification has a bad correlation with surgical complications. If they opt by using ENZIAN classification, how they will built the final ENZIAN score?. Finally, although other definitions should be also provided, one the most important missing definition is the core of this protocol: how the author will define “surgical complication” and how they will define “major and minor complications”?. Other important missing definitions are how they will define “level care of centers”, “expertise of the surgery team”… Furthermore, the authors should explain why they are not taking in account other confounding factors as the association with adenomyosis is or if they will consider as a complication of endometriosis surgery the impairment of ovarian function frequently seen after ovarian endometrioma stripping.

ANSWER: We will conduct subgroup analysis by indication of surgeries (infertility, or pain) which are reported in most studies. However, we will not analyze data regarding fertility rates or pain following surgery as outcomes. Based on studies' reports, endometriosis severity is categorized by rASRM or/and ENZIAN. Gynecological experts in our team will be consulted if necessary to dichotomize studies in two groups according to these variables and depending on available data. Any ambiguity in the reported data will also be referred to the authors of the original studies. We will define surgical complications as any complications (composite outcome) which occur during surgery (intraoperative complications) and after the surgery (post-operative complications). Furthermore, a post-operative complication related to surgery generally occurs shortly after the procedure. Therefore, after consulting our expert, we remove complications occurring after one year from the manuscript. Therefore, postoperative complications are reported at 30 days and at three months following surgery when possible. You will find the list of complication as a table below. The symptoms of endometriosis after the surgery such as pain or fertility rate are not considered as part of postoperative complications in this study.

Regarding the major or minor complications, we use the Clavien–Dindo classification (grade I-II: minor complications, grade III-IV: major complications, grade V: death). This classification relies on grading complications based on the therapy used to treat the complication. A gynecological expert in our team will be consulted is it is necessary For the level of care, we will classify them as rural/community/regional hospitals versus academic/national/specialized hospitals versus unknown or mixed hospitals. 

We planned to extract expertise of surgeons from the included articles when available. However, we expect there will be a high degree of heterogeneity in the way it is reported, and no official classification is available. Hence, we will report this information as part of descriptive analyses, or will create categories if deemed appropriate after consultation with clinicians and method experts in our group. The manuscript has been modified accordingly. The adenomyosis information will be collected if it is available in the studies for subgroup analysis. Regarding the ovarian reserve following surgical excision of ovarian endometriomas, as mentioned before we will not consider fertility rates as an outcome in this study.

Line 147-155: “[Our primary outcome will be a composite outcome of any intraoperative and postoperative complications (including conditions such as hemorrhage; organ injury; nerve damages including urinary voiding dysfunction, and urinary retention; open surgery conversion; mortality; postoperative reintervention; readmission after discharge from hospital; length of surgery; and length of hospital stay). The full list of complications is provided in the S2 Appendix. In addition, We will categorize complications as intraoperative complications, which occur during surgery, and postoperative complications, which occur after surgery. Postoperative complications will be reported within 30 days and three months after the surgery.]”

Line 165-175: “[The severity of endometriosis will be classified according to the revised American Society for Reproductive Medicine (rASRM) or/and ENZIAN. In addition, the level of care will be classified as rural/community/regional hospitals versus academic/national/specialized hospitals versus unknown or mixed hospitals. When surgeon expertise is available in the included articles, we will extract it. Since there is no official classification, we will create categories, if necessary, after consulting with clinicians and method experts in our group, unless we report it as a part of descriptive analyses. Furthermore, the Clavien-Dindo classification is used to classify major and minor complications (grades I-II: minor complications, grades III-IV: major complications, grade V: death). This classification determines the severity of complications by analyzing the therapy used to treat them. If necessary, we will consult a gynecological expert in our team.]”

Line 194-195: “[ The adenomyosis information also will be collected if it is available in the studies for subgroup analysis.]” 

Reviewer #2: Dear

Thank you to giving me the chance to review this interesting protocol on a very important topic for the management of endometriosis. In order to explain better to the readers the importance of some important factors influencing iatrogenic complications (ie ureteral stent, parametrial or vaginal mucosa Involvement) related to deep infiltrating endometriosis surgery I suggest to include and briefly comments some relevant articles (eg doi 10.1002/ijgo.14089, 10.1002/ijgo.13959, 10.1111/aogs.13824).

I believe that is important to list exauatively all complications that will be assessed, both de novo functional than organic ones. 

ANSWER: The table below, which contain the list of complications, is now provided in an appendix (S2) as a part of the data extraction form.

Composite outcome

Intraoperative complication

Post-operative complications (overall)

Post-operative complications in first month

Post-operative complications in three months

Grade I-II CD (minor complication)*

Grade III- V DC (major complications) *

Length of surgery 

Readmission

Length of hospitalization

Open surgery conversion

Post-operative reintervention 

 Laparoscopic revision

 laparotomy

Infection 

Wound infection

Urinary tract infection

Vaginal infection

Abscess

Pyelonephritis

Cellulitis

Sepsis

Fever/pyrexia

Vascular repair

Venous thromboembolism

Thrombophlebitis

Pulmonary embolus

pneumothorax

Damage to blood vessels

Genitourinary tract injury

Cystotomy

Ureteral injury (transection)

Vesicovaginal fistula

Ureterovaginal fistula

Urinary retention

Bladder atony

voiding dysfunction

self-catheterization (< and > 30 days)

Gastrointestinal tract injury

Injury to gastrointestinal tract

Anastomotic dehiscence/anastomotic leak syndrome

Rectovaginal fistula

Rectal stenosis

Rectal perforation

ileostomy

Bleeding

Blood transfusion

Hematoma

Vascular injury

Vaginal cuff dehiscence

Nerve injury

Repair to bladder

Repair to ureter

Repair to bowel

Other complications¶

* Grade I: was assigned to any deviation from the normal postoperative course without the need for pharmacological treatment or surgical, endoscopic, and radiological interventions. Allowed therapeutic regimens are drugs as antiemetics, antipyretics, analgetics, diuretics and electrolytes, and physiotherapy. This grade also includes wound infections opened at the bedside;

Grade II: Requiring pharmacological treatment with drugs other than such allowed for Grade I complications. Blood transfusions and total parenteral nutrition are also included; 

Grade III: requiring surgical, endoscopic, or radiological intervention; 

Grade III-a: intervention not under general anesthesia; 

Grade III-b: intervention under general anesthesia; 

Grade IV: life-threatening complication (including CNS complications) requiring IC/ICU-management; 

Grade IV-a: single organ dysfunction Grade IV-b: multi organ dysfunction; 

Grade V: death of a patient.

¶ Other complications included accidental cut, puncture, perforation, or hemorrhage during medical care and surgical procedures as the cause of abnormal reaction of patient or later complication

Line 74-76: “[In addition, iatrogenic pelvic organ dysfunction can happen due to accidental damage to pelvic nerves in complete excision of deep-infiltrating endometriosis despite practices in nerve-sparing surgeries.]”

---

## [Decision Letter · Decision Letter 1]

5 May 2023

Complications following surgeries for endometriosis: a systematic review protocol

PONE-D-22-21731R1

Dear Dr. Boutin,

We’re pleased to inform you that your manuscript has been judged scientifically suitable for publication and will be formally accepted for publication once it meets all outstanding technical requirements.

Kind regards,

Diego Raimondo

Academic Editor

PLOS ONE

Additional Editor Comments (optional):

Reviewers' comments:

Reviewer's Responses to Questions

**Comments to the Author**

1. Does the manuscript provide a valid rationale for the proposed study, with clearly identified and justified research questions?

Reviewer #1: Yes

2. Is the protocol technically sound and planned in a manner that will lead to a meaningful outcome and allow testing the stated hypotheses?

Reviewer #1: Yes

3. Is the methodology feasible and described in sufficient detail to allow the work to be replicable?

Reviewer #1: Yes

4. Have the authors described where all data underlying the findings will be made available when the study is complete?

Reviewer #1: Yes

5. Is the manuscript presented in an intelligible fashion and written in standard English?

Reviewer #1: Yes

6. Review Comments to the Author

You may also provide optional suggestions and comments to authors that they might find helpful in planning their study.

Reviewer #1: The questions raised by this reviewer have been answered and the paper is now worthy of publication.

7. PLOS authors have the option to publish the peer review history of their article (what does this mean?). If published, this will include your full peer review and any attached files.

Reviewer #1: No

---

## [Editor Report · Acceptance letter]

15 May 2023

PONE-D-22-21731R1 

Complications following surgeries for endometriosis: a systematic review protocol 

Dear Dr. Boutin:

I'm pleased to inform you that your manuscript has been deemed suitable for publication in PLOS ONE. Congratulations! Your manuscript is now with our production department. 

Kind regards, 

on behalf of

Dr. Diego Raimondo 

Academic Editor

PLOS ONE